# Polymeric Films Containing Tenoxicam as Prospective Transdermal Drug Delivery Systems: Design and Characterization

**Adriana Ciurba** [1], **Paula Antonoaea** [1,*], **Nicoleta Todoran** [1], **Emőke Rédai** [1], **Robert Alexandru Vlad** [1], **Anamaria Tătaru** [1], **Daniela-Lucia Muntean** [2] **and Magdalena Bîrsan** [3]

1   Department of Pharmaceutical Technology and Cosmetology, Faculty of Pharmacy, University of Medicine, Pharmacy, Science and Technology, "George Emil Palade" of Targu Mures, 38 Gheorghe Marinescu Street, 540142 Targu Mures, Romania; adriana.ciurba@umfst.ro (A.C.); nicoleta.todoran@umfst.ro (N.T.); emoke.redai@umfst.ro (E.R.); robert.vlad@umfst.ro (R.A.V.); farmacist.anamariatataru@gmail.com (A.T.)
2   Department of Analytical Chemistry and Drug Analysis, Faculty of Pharmacy, University of Medicine, Pharmacy, Science and Technology "George Emil Palade" of Targu Mures, 38 Gheorghe Marinescu Street, 540142 Targu Mures, Romania; daniela.muntean@umfst.ro
3   Department of Pharmaceutical Technology, Faculty of Pharmacy, "Grigore T. Popa" University of Medicine and Pharmacy of Iasi, 16 Universitatii Street, 700115 Iasi, Romania; magdalena.birsan@umfiasi.ro
*   Correspondence: paula.antonoaea@umfst.ro; Tel.: +40-747-693-856

**Abstract:** The administration of drugs via transdermal therapeutic systems has become an attractive form of therapeutic approach, considering its advantages and the high patient compliance achieved, making them a viable alternative, especially in the treatment of chronic diseases. The purpose of our study was the development of polymer-based films containing tenoxicam (TX) and the analysis of dissolution kinetics. Auxiliary substances represent an important part of pharmaceutical forms, so during the first stage, TX and excipient compatibility were verified. Fourier Transform Infrared Spectroscopy (FT-IR) and Differential Scanning Calorimetry (DSC) analyses were performed on TX and on physical mixtures of TX-HPMC$_{E5}$ and TX-HPMC$_{15kcP}$. Three polymeric films of TX (TX$_1$, TX$_2$, and TX$_3$) were prepared using a solvent evaporation technique. Release studies were done at 32 °C ± 1 °C with a Franz diffusion cell. The results of the DSC and FT-IR analyses demonstrated the compatibility of the active substance with the two matrix-forming polymers. The results obtained in the release studies of TX from the proposed polymeric films suggested a pH-dependent behavior in all three polymeric films. At pH 5.5, flux values were between $8.058 \pm 0.125$ µg·cm$^{-2}$·h$^{-1}$ and $10.850 \pm 0.380$ µg·cm$^{-2}$·h$^{-1}$; and at pH 7.4, between $10.990 \pm 0.2.490$ µg·cm$^{-2}$·h$^{-1}$ and $53.140 \pm 0.196$ µg·cm$^{-2}$·h$^{-1}$. The Korsmeyer–Peppas model described a non-Fickian transport mechanism. The *n* values varied between 0.63–0.7 at pH 5.5 and 0.73–0.86 at pH 7.4, which suggested a diffusion depending on the matrix hydration and polymer relaxation.

**Keywords:** tenoxicam; polymeric film; DSC; FT-IR; kinetic analysis

## 1. Introduction

The physicochemical properties of active pharmaceutical ingredients lead to permeation of the skin barrier, and these ingredients are added to the barrier to external factors that the skin naturally manifests. This barrier, the stratum corneum, is the outermost layer of the skin, and is constituted of enucleated cells known as corneocytes. The formation of these cells occurs through the organization of globular proteins liberated by keratohyalin granules around intermediate keratin filaments. The stratum corneum contains about 40% proteins (mostly keratin), 40% water, and 15–20% lipids (15% triglycerides and free fatty acids, 25% cholesterol, 50% sphingomyelin, and other lipids in smaller amounts). Ceramides, which belong to the class of sphingolipids, are responsible for forming structures

capable of water fixation on their hydrophilic regions. These structures encircle corneocytes, acting as an impermeable barrier with a natural moisturizing factor (NMF). They contain lipids, hydroxyl acids, amino acids, urea, and inorganic acids. Keratocytes act as an impermeable membrane and limit water loss. Lipids interfere with the regulation mechanisms of the transepidermal penetration of hydrophobic substances. Water is retained in the stratum corneum, serving as a plasticizer. Lipids are mostly located extracellularly, and in proteins, both intra- and extracellularly. The lipid content of the stratum corneum provides a low permeability for many external agents, and at the same time protects against them [1–5].

The administration of drugs via transdermal therapeutic systems has become an attractive form of therapeutic approach, considering its advantages and the high patient compliance achieved, making them a viable alternative, especially in the treatment of chronic diseases. Formulation of the new polymer-based film requires the optimization of the active drug amount in the skin in order to avoid supersaturation, depending mostly on the physicochemical properties of the drug (molecular weight, repartition coefficient, solubility) and optimization of the pharmaceutical form. The film must be nonirritating and must adhere properly to the administration zone and the occlusive folia. Another property that must be realized is ease of removal of protective folia [6–8].

The first phase in the formulation of polymer-based transdermal films is the selection of active substances and excipients, followed by obtaining and evaluating the product. Using in vitro and in vivo studies, a proper transdermal flux must be achieved to ensure the correct daily dose of active drug. In the second phase, manufacturing aspects are considered in order to ensure drug stability, low costs, and patient comfort.

Cellulose ether-type matrix-forming polymers control the release rate of the active drug from the transdermal therapeutic system. Drug–polymer compatibility and a lack of unwanted interactions between the active substance and auxiliary materials is required [9–11].

A polymer-based transdermal film usually contains permeation enhancers, which, by interacting with the structural components of the stratum corneum, enhance the permeation of the active ingredients through the skin [12]. The role of surfactants in the composition of dermal preparation is given by their amphiphilic structure that enhances the permeation for the active ingredient [13]. For example, an in vivo study on mouse skin showed an improvement in the permeation enhancement by Tween 20 for hydrocortisone and lidocaine [14,15]. Surfactants with a low molecular weight in contact with skin act like permeation enhancers by using mechanisms such as protein binding to the skin surface, denaturation of proteins, solubilization of intercellular lipids of the stratum corneum, or penetration of the epidermal lipid barrier [16].

Tween 20 (Tw$_{20}$), or Polysorbatum 20, 2-[2-[3,4–bis (2-hydroxyethoxy) oxolan-2-yl]-2-(2-hydroxyethoxy) ethoxy] ethyl dodecanoate, has an Hydrophilic-lipophilic balance (HLB) of 16.7, and is soluble in water, ethanol, methanol, and ethyl-acetate [17]. Non-ionic surfactants such as propylene glycol are less toxic and irritating, making them eligible for use in polymeric films [13,18]. Propylene glycol, or propane-1,2-diol, is a viscous, colorless liquid that is nearly odorless, and has a sweet taste. It is miscible with water and organic solvents (alcohol, chloroform, and acetone). The US Food and Drug Administration [19] classifies propylene glycol as safe for alimentation, cosmetics, and pharmaceuticals. In transdermal therapeutic systems, it can be used as a plasticizer and permeation enhancer. Literature data [20,21] confirms that permeation enhancement takes place using various mechanisms, including competition with water molecules for hydrogen bonding, keratin hydration in the stratum corneum, and intercalation of propylene glycol molecules between the polar groups of bilayered lipids, which enhances the transdermal permeation of lipophilic drugs.

Tenoxicam (TX) is a nonsteroidal anti-inflammatory drug (NSAID), 4-hydroxy-2-methylthieno[2,3-e][1,2]thiazine-3-carboxylic acid 1,1-dioxide, that is practically insoluble in water (14.1 μg/mL), freely soluble in ethanol, log P: 1.9 in water/octanol, 0.3 at pH 7.4 and 3.5 at pH 2.1. The side-effect profile of TX resembles the profile of other NSAIDs. In addition to enhancing patient compliance by reducing the number of dose administrations

and minimizing gastric, hepatic, and renal side effects, transdermal administration is more suitable in cases of oral intolerance [22–25]. In a study published by Nessem et al., transdermal films with TX were developed in order to reduce the side effects of the drug. The in vitro release studies were conducted using a cellophane dialysis membrane in a phosphate buffer at pH 7.4 as a receptor medium. Depending on the composition of the formulations, the percent of drug released varied between 40.87% and 80.89%. Another study published by Ramkanth et al. in 2015 revealed the formulation and evaluation of patches containing TX. The in vitro drug studies indicated a cumulative percentage of drug released in 24 h at 99.28% in a phosphate buffer at pH 7.4 [24,26]. Auxiliary substances represent an important part of a pharmaceutical formulation, because in the first stage, TX and excipient compatibility should be verified. Although an unanimously accepted protocol does not exist to evaluate the incompatibility of auxiliary substances in drugs, multiple studies have reported thermal and non-thermal analytical methods for auxiliary substance selection. Widely used screening methods include differential scanning calorimetry (DSC) and Fourier transformed infrared spectrometry (FT-IR).

DSC, which is a method of evaluating the incompatibilities of associated components in one pharmaceutical form, presents multiple advantages, including small sample size, short determination time, and wide temperature range ($-120/600$ °C) [27]. Modification or disappearance of an endo- or exothermal peak indicates a possible incompatibility. Frequently, a thermal shift of a characteristic peak may be observed, explained by strong interactions among drug and excipient, but this is not necessarily an incompatibility [28,29]. This explains the necessity of acquiring thermal behavior data with alternative methods, preferably non-thermal, such as Fourier-transform infrared spectrometry [30].

FT-IR is usually used as a complementary method to confirm DSC data [31]. The appearance of new absorption bands or modification of existing ones indicates a potential interaction among the associated drug and the auxiliary substances.

The solvent-casting method to obtain a hydrophilic matrix is frequently used in experimental studies. The aim of this work was the development of tenoxicam-containing polymer-based films and the analysis of dissolution kinetics. Dissolution studies were performed using the stationary Franz cell method. The Franz cell diffusion system, which has applicability in dermal permeation studies of active substances, provides relevant information on the formulation and on skin behavior. The diffusion of a drug from transdermal therapeutic systems is a complex process characterized by the mobilization of drug molecules among the polymeric chains of the matrix in contact with a membrane. These passive phenomena are followed by the release of the drug through the membrane, known as the donor phase, and dissolution in the receptor solution for quantification.

## 2. Materials and Methods

### 2.1. Materials

The tenoxicam was purchased from Nantong Chemding Chephar Co. Ltd. (Jiangsu, China). The hydroxypropyl methylcellulose 15kcP was purchased from Shin-Etsu Chemical Co., Ltd. (Tokyo, Japan), and the hydroxypropyl methylcellulose E5 was purchased from Dow Chemical Co. (Midland, TX, USA). The propylene glycol was purchased from Scharlau Chemie (Barcelona, Spain), and the Tween 20 was purchased from Sigma Aldrich (Milano, Italy). All other reagents were of analytical grade. According to the USP 29 specification of US Pharmacopeia and other literature sources, hydroxypropyl methylcellulose E5 ($HPMC_{E5}$) contains 28–30% methoxide groups ($-OCH_3$) and 7–12% hydroxyl-propyl groups ($-OCH_2CHOHCH_3$). A 2% solution presents a viscosity of de 4–6 mPa·s (cP). Hydroxypropyl methylcellulose 15kcP ($HPMC_{15kcP}$) contains 19–24% methoxide groups ($-OCH_3$) and 4–12% hydroxypropyl groups ($-OCH_2CHOHCH_3$). A 2% solution presents a viscosity of de 15,000 mPa·s (cP)



## 2.2. Methods

### 2.2.1. Fourier Transform Infrared Spectroscopy (FT-IR)

FT-IR analysis was performed on the TX and physical mixtures of TX-HPMC$_{E5}$ (1:1) and TX-HPMC$_{15kcP}$ (1:1). A total of 150 mg KBr and 1 mg of each sample were mixed and compressed using a hydraulic press. The obtained pellets were scanned in the following experimental conditions: spectral-domain 400–4000 cm$^{-1}$; resolution of 4 cm$^{-1}$; eight scans. The FT-IR spectra of the samples were collected using an FT-IR Spectrometer Thermo-Nicolet, an Avatar 330, and Omnic 10.1 software. The FT-IR studies were done at room temperature (25 °C ± 2 °C). This method was used to conduct a preliminary study on the development of polymeric films with TX. This method did not involve thermal stress on the samples, and a physical–chemical modification did not occur during spectrometry.

### 2.2.2. Differential Scanning Calorimetry (DSC)

The DSC analysis was performed on the TX and physical mixtures of TX-HPMC$_{E5}$ (1:1) and TX-HPMC$_{15kcP}$ (1:1). The DSC experiments were performed on 5 mg samples that were placed in aluminum pans (40 μL). The rate of heating was 5 °C min$^{-1}$ in a temperature range of 30–300 °C. The DSC thermograms were recorded using the Shimadzu TA-60WS differential scanning calorimeter, with TA-60 software. The DSC method was used as a preliminary study in the development of polymeric films with TX. The comparative analysis of registered thermic phenomena for the TX and binary mixtures of film-forming polymer and TX generated pertinent information regarding compatibility.

### 2.2.3. Polymeric Film Preparation

Three polymeric films (Table 1) were prepared using the solvent evaporation technique. The drug was dispersed under continuous stirring (500 rpm, Heidolph RYR1 Stirrer, Germany) in a mixture of alcohol—propylene glycol. Tw$_{20}$ was dissolved into the corresponding amount of water and then added to the alcoholic solution of TX. The polymer was dispersed in the resulting solution and mixed for 1 h. The obtained polymeric solutions were left in the ultrasonic water bath for another hour until the air bubbles disappeared. A total of 20 g of each polymeric composition was poured into Petri glasses with a diameter of 9.8 cm. The obtained films were dried at 40 °C in a hot-air oven for 24 h. Following the drying process, the polymeric films were wrapped in aluminum foil and kept at 20 °C ± 5 °C for further analysis.

**Table 1.** Composition of the formulations TX$_1$, TX$_2$, and TX$_3$.

|  |  | **TX$_1$** | **TX$_2$** | **TX$_3$** |
|---|---|---|---|---|
| Tenoxicam | drug | 0.5 | 0.5 | 0.5 |
| HPMC$_{E5}$ | film former polymers | 3.0 | - | - |
| HPMC$_{15kcP}$ |  | - | 1.0 | 1.5 |
| Tween 20 | permeation enhancer | 1.0 | 1.0 | 1.0 |
| Propylene glycol | permeation enhancer/plasticizer | 10.0 | 10.0 | 10.0 |
| Alcohol 96° | solvents | 30.0 | 30.0 | 30.0 |
| Ultrapure water |  | 55.5 | 57.5 | 57.0 |

### 2.2.4. Evaluation of Dissolution Kinetics from Polymeric Films

The release studies were done at 32 °C ± 1 °C using a Franz diffusion cell (Hanson Research). The receptor compartment was filled with 14 mL of air-free bubble solution and stirred with a magnetic bead at 300 rpm. A phosphate buffer at pH 5.5 and a phosphate buffer at pH 7.4 were used as receptor solutions. A nylon-type membrane with a diameter of 25 mm and a thickness of less than 1 mm was placed between the receptor and donor compartment. To ensure the adhesion of the polymeric film, 200 μL of phosphate buffer

was placed over the membrane and was left for 1 h in order to moisturize. Finally, a polymeric film circle with a diameter of 1.8 cm (2.54 cm$^2$ surface) was placed on the donor compartment and covered with parafilm. In the dissolution studies, the samples of the polymeric films had a weight of 85.17 $\pm$ 0.06 mg and a theoretical TX concentration of 3.37 mg. Tests on samples of 1 mL were carried out for 30 h. The same volume of fresh phosphate buffer maintained at 32 °C $\pm$ 1 °C was added to the receptor compartment after sampling. The TX concentration in the samples was analyzed at 360 nm using a validated HPLC method [32].

### 2.2.5. Kinetic Analysis of the Release Profiles

In order to compare the release rate of the TX from the polymeric films, the parameter area under the curve (AUC) was calculated and analyzed. Mathematical modeling of the dissolution curves using mathematical functions describing five different dissolution kinetics (Higuchi with $T_{lag}$; Higuchi with $F_0$; Korsmeyer–Peppas; Korsmeyer–Peppas with $T_{lag}$; and Korsmeyer–Peppas with $F_0$) was realized by graphical simulation curves for every model. For every mathematical model, two curves were compared: the real curve (the experimental curve) and the predicted curve (obtained using software) [33,34]. The software used was the DDSolver Add-In for Microsoft Excel. The calculated kinetic parameters were: best of fit values and goodness of fit values.

## 3. Results and Discussion

### 3.1. FT-IR Analysis

In the IR spectrum of TX (Figure 1), the following characteristic peaks in good correlation with literature data were identified: 3433.61 cm$^{-1}$ (stretching of –OH), 3119.76 cm$^{-1}$ and 3091.33 cm$^{-1}$ (stretching of C-H and N-H groups), 1637.40 cm$^{-1}$ (stretching of CO-NH), 1597.95 cm$^{-1}$ (stretching of C=N), 1327.54 cm$^{-1}$ (asymmetric stretching of m(SO$_2$)) and 1042.9 cm$^{-1}$ (symmetric stretching of (SO$_2$)) [35–37]. The IR spectrum of the proposed mixtures (Figure 2) showed the upshifted characteristic peaks of TX. The lack of significant modification in the spectra confirmed the lack of interaction among TX and the excipients. This assumption was also confirmed by the published data, which show the lack of any interaction of TX with polymers like hydroxypropyl methylcellulose. Similar results were published in 2015 by Ramkanth et al. [24] concerning the incompatibilities of TX in hydroxypropyl methylcellulose (HPMC)-based films.

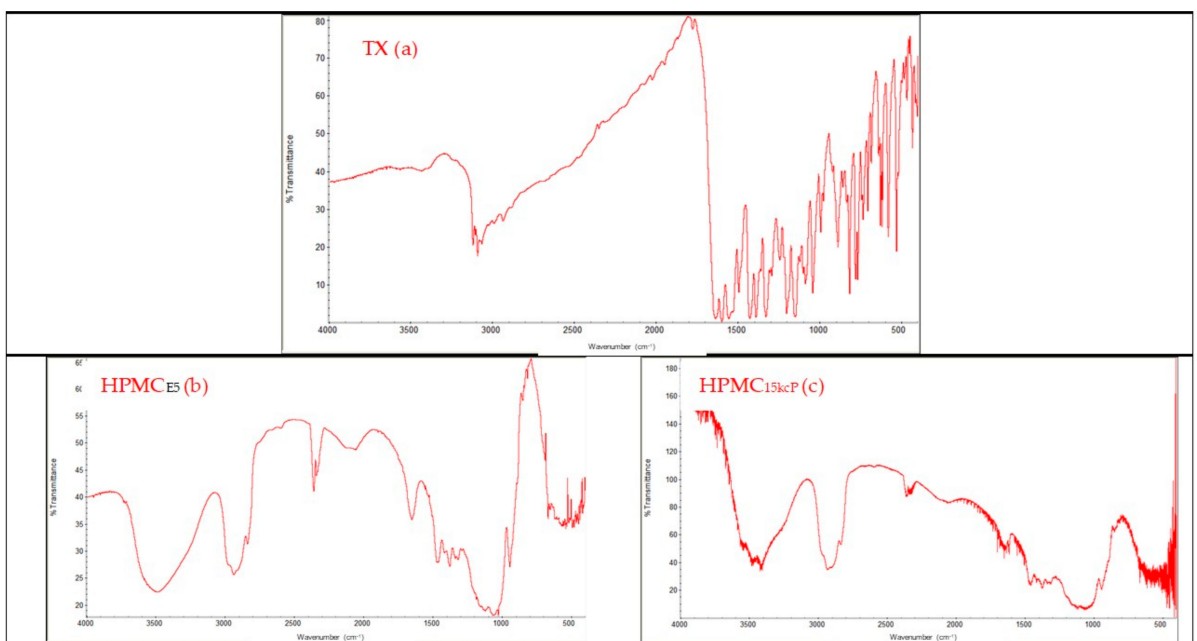

**Figure 1.** The FT-IR spectra of: TX (**a**), HPMC$_{E5}$ (**b**), and HPMC$_{15kcP}$ (**c**).

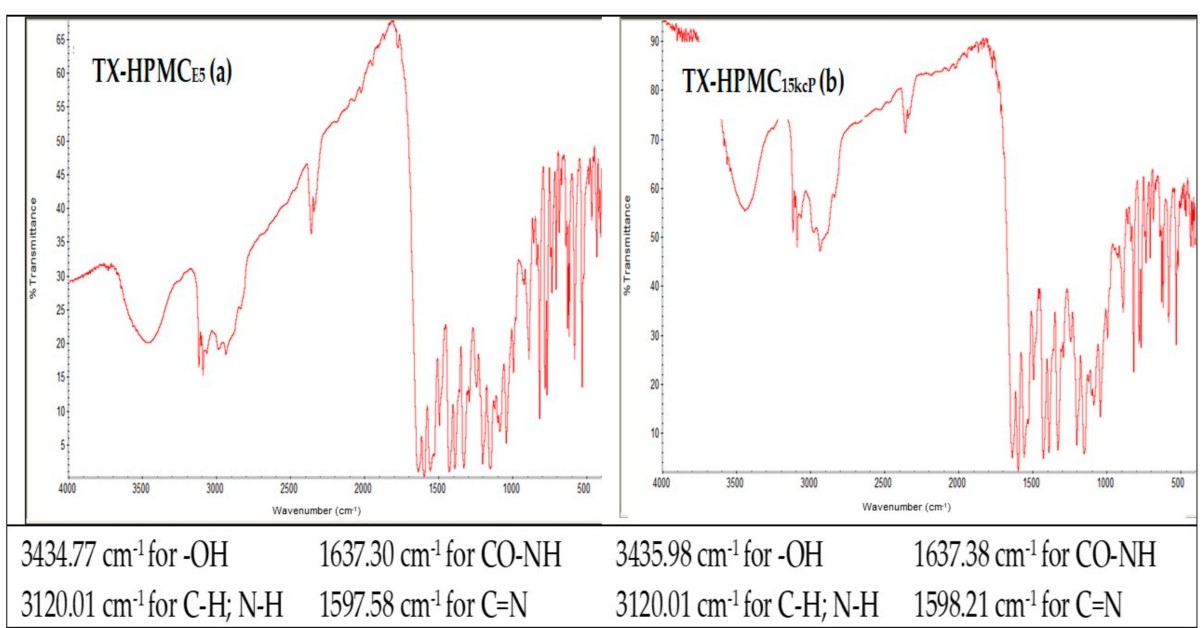

| 3434.77 cm⁻¹ for -OH | 1637.30 cm⁻¹ for CO-NH | 3435.98 cm⁻¹ for -OH | 1637.38 cm⁻¹ for CO-NH |
| 3120.01 cm⁻¹ for C-H; N-H | 1597.58 cm⁻¹ for C=N | 3120.01 cm⁻¹ for C-H; N-H | 1598.21 cm⁻¹ for C=N |

**Figure 2.** The FT-IR spectra of physical mixtures: TX-HPMC$_{E5}$ (**a**) and TX-HPMC$_{15kcP}$ (**b**).

### 3.2. DSC Analysis

The DSC curve of TX (Figure 3) shows an endothermic peak indicating melting of TX at 217.15 °C, immediately followed by an exothermic decomposition at 219.70 °C. In the DSC thermograms of HPMC$_{E5}$ and HPMC$_{15kcP}$, we can observe an endothermic peak between 50–100 °C. The thermal behavior of the two polymers can be attributed to their dehydration due to the –OH groups in their chemical structure, followed by their decomposition. In thermograms, the binary mixtures of TX and the studied polymers indicated the characteristic comportment of TX. The curve of TX-HPMC$_{E5}$ showed an exothermic peak at 207.51 °C, while TX-HPMC$_{15kcP}$ presented a peak at 211.32 °C. In binary mixtures, the weight ratio of active substance and polymer used (2.5 mg:2.5 mg) influenced the peak temperatures, and thermal shifts of the peaks could also be explained by the recognition of polymer as an impurity by the active substance [38]. Even if the thermal behavior of the physical mixture could be associated with the interactions between the TX and the two polymers, it cannot be stated with certainty. The peak deviations of the TX in DSC thermograms of the analyzed physical mixtures can be explained by the changes in some of its properties, such as the transition from the crystalline form to the amorphous form, an assumption supported by the data from the literature [29].

### 3.3. Polymeric Films Characterization

We obtained three polymeric films with a surface of 75.39 cm$^2$ surface, a weight of 2.528 ± 0.06 g, and a theoretical TX concentration of 1.326 mg/cm$^2$.

### 3.4. Evaluation of Dissolution Kinetics from Polymeric Films

The results obtained in the release studies of the TX from the proposed polymeric films are presented in Figures 4 and 5. A pH-dependent behavior was observed in all three polymeric films. Regardless of the matrix type, the dissolved amount of TX was 2–4 times higher at pH 7.4 compared to the results achieved at pH 5.5, suggesting possible difficulties of the TX in transiting the stratum corneum. After 30 h at pH 7.4, the dissolved TX concentrations (TX$_1$: 79.77%; TX$_2$: 95.84%; TX$_3$: 54.57%) confirmed that an optimized formulation could achieve therapeutic concentration for a long period of time. At pH 5.5, no significant differences were observed between the dissolved TX concentrations from TX$_1$ and TX$_2$. However, TX$_3$ formulation liberated 8% more active substance, suggesting the discriminative role of HPMC$_{15kcP}$ concentration. Dissolution at pH 7.4 differentiated

the characteristics of the three polymeric films to a greater degree. From $TX_1$, based on low viscosity polymer, 79.77% TX was dissolved; from $TX_2$ containing 1% high viscosity polymer, 25% more TX was liberated compared to $TX_1$. When the polymer concentration was increased from 0.5% to 1.5% $HPMC_{15kcP}$ in formula $TX_3$, a significant decrease in the amount of TX amount was detected (40%). The higher viscosity of polymeric film $TX_3$ could be correlated with the mobilization difficulty of active substance molecules from the polymer chains. The obtained results can be attributed to the fact that the acidic pH caused an increase in the number of electrically charged molecules. This behavior could explain the decrease in passive diffusion capacity under the action of the concentration gradient generated on both sides of the membrane.

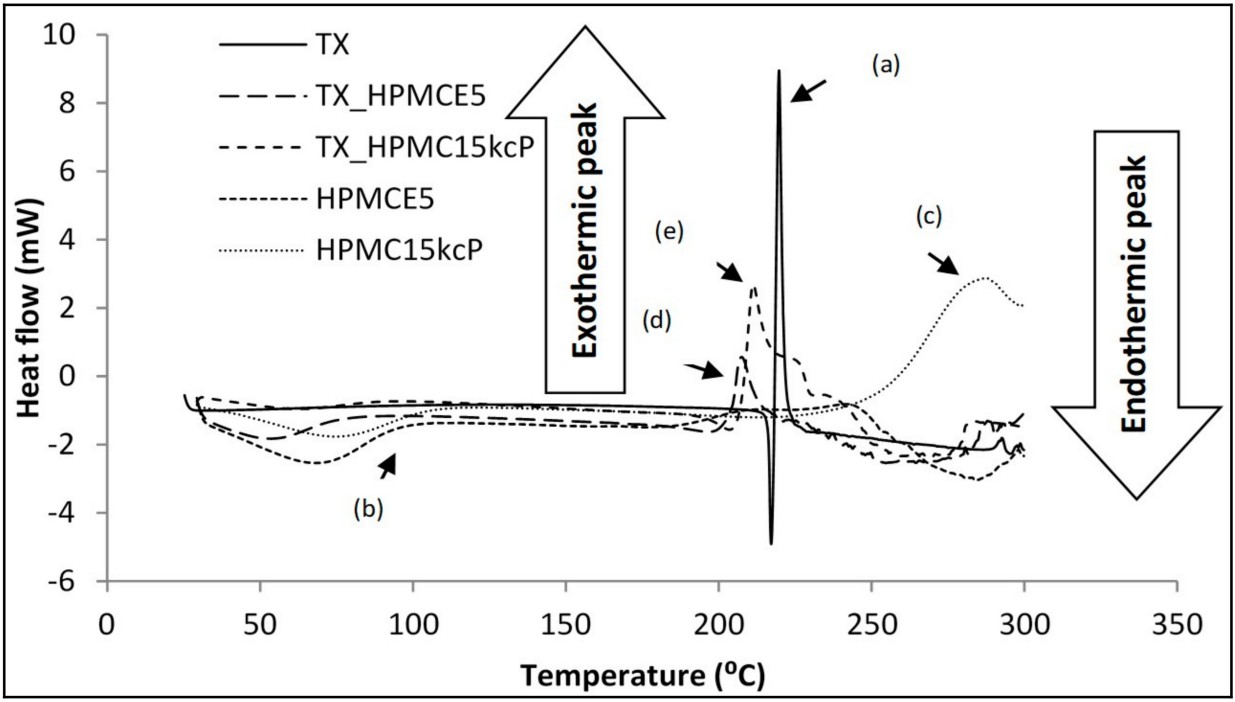

**Figure 3.** The differential scanning calorimetry thermogram of TX (**a**), $HPMC_{E5}$ (**b**), $HPMC_{15kcP}$ (**c**), and physical mixtures TX-$HPMC_{E5}$ (**d**) and TX-$HPMC_{15kcP}$ (**e**).

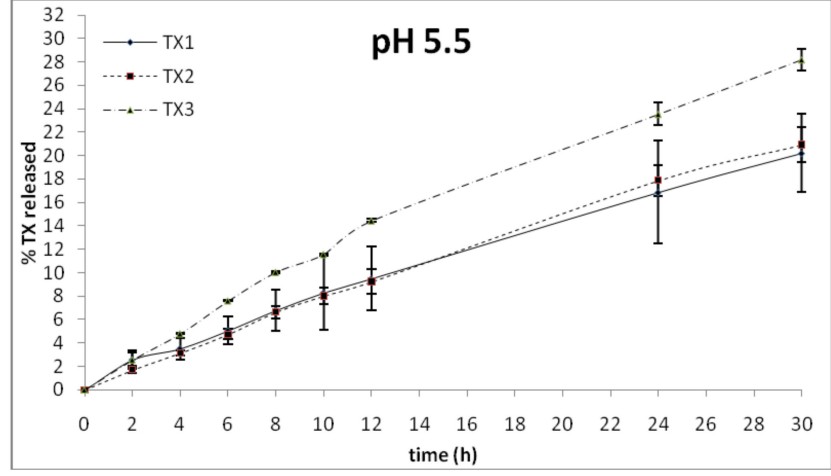

**Figure 4.** TX release profiles at pH 5.5.

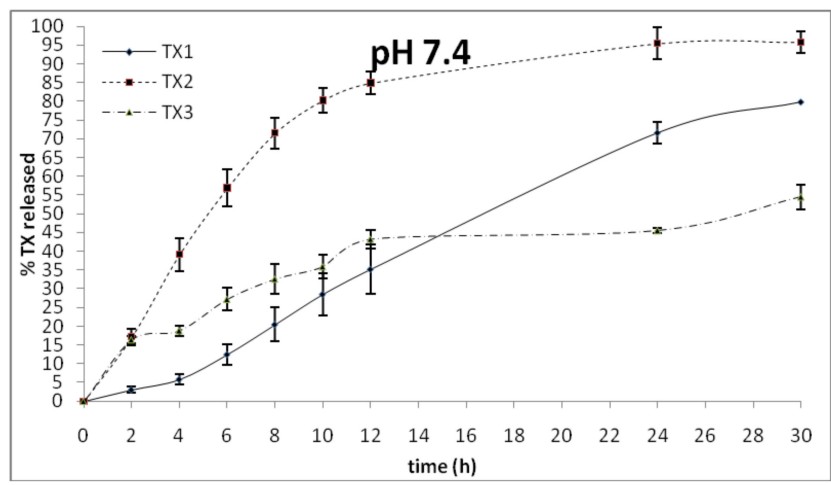

**Figure 5.** The TX release profiles at pH 7.4.

The Dissolution curve analysis was based on the area under the curve (AUC) values, depending on the polymer matrix.

The AUC values (Table 2 and Figure 6) indicated a major influence of the acceptor medium pH on the dissolution rate. Formulation $TX_1$ presented a dissolution rate four times higher at pH 7.4 compared to pH 5.5. At pH 7.4, the dissolution rate increased almost sevenfold for the $TX_2$ formulation and threefold for $TX_3$. At pH 5.5, AUC values showed an increasing dissolution rate in the order $TX_1 < TX_2 < TX_3$, indicating the polymer type and the concentration as influencing parameters. Similar behavior was observed at pH 7.4 for formulations $TX_1$ and $TX_2$. The AUC value for $TX_3$ indicated a different dissolution, correlated with the high-viscosity polymer matrix and a higher concentration of polymer.

**Table 2.** AUC values.

| AUC | pH 5.5 | pH 7.4 |
|-----|--------|--------|
| $TX_1$ | 331 | 1270 |
| $TX_2$ | 339 | 2269 |
| $TX_3$ | 471 | 1137 |

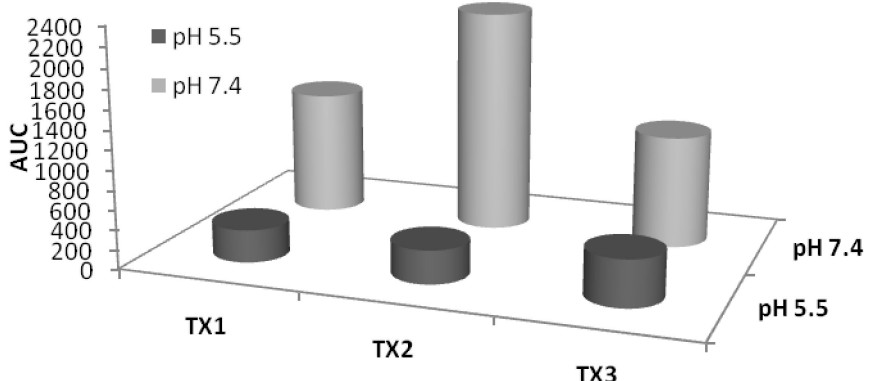

**Figure 6.** The influence of the polymeric films, depending on pH, on the dissolution rate.

3.4.1. Evaluation of Dissolution Kinetics from Polymer Films

The flux of drug calculations were realized using simple regression of the steady-state portion of dissolution curves. In other words, this parameter was calculated for the time interval (h) between $t_{initial}$ and $t_{final}$ for the corresponding concentrations of TX ($C_{initial}$ and $C_{final}$, respectively—% TX dissolved), and the slope of the regression curve was represented.

The cumulative amounts of the assayed active substance at the mentioned time points are presented in Table 3.

**Table 3.** The kinetic parameters for TX.

| pH | pH 5.5 | | | pH 7.4 | | |
|---|---|---|---|---|---|---|
| **Polymeric Film** | **$TX_1$** | **$TX_2$** | **$TX_3$** | **$TX_1$** | **$TX_2$** | **$TX_3$** |
| **Linear regression/ Flow: J $\pm$ DS ($\mu$g·cm$^{-2}$·h$^{-1}$)** | 8.058 $\pm$0.125 | 9.345 $\pm$0.135 | 10.850 $\pm$0.380 | 53.140 $\pm$0.196 | 13.410 $\pm$2.313 | 10.990 $\pm$2.490 |
| **Steady-state: calculated parameters** | | | | | | |
| $t_{initial}$ (h) | 8.02 | 8.02 | 8.02 | 6.06 | 4.10 | 4.10 |
| $c_{initial}$ % $\pm$ DS | 6.99 $\pm$0.37 | 6.58 $\pm$0.35 | 10.37 $\pm$1.14 | 12.70 $\pm$0.47 | 40.48 $\pm$3.96 | 19.67 $\pm$3.72 |
| $t_{final}$ (h) | 30 | 23.98 | 30 | 9.98 | 7.88 | 7.88 |
| $c_{final}$ % $\pm$ DS | 20.34 $\pm$0.47 | 17.82 $\pm$0.52 | 28.35 $\pm$1.47 | 28.41 $\pm$0.48 | 71.04 $\pm$3.9 | 32.64 $\pm$2.6 |
| $t_{final}$—$t_{initial}$ (h) | 21.98 | 15.96 | 21.98 | 3.92 | 3.78 | 3.78 |
| $R^2$ | 0.9993 | 0.9996 | 0.9962 | 1.000 | 0.9971 | 0.9842 |
| **Kinetic modelling on TX release from the polymeric film** | | | | | | |
| **Higuchi with $T_{lag}$: F = $k_H$*(t-$Tl_{ag}$)^0.5** | | | | | | |
| $k_H$ | 47.729 | 50.346 | 67.307 | 200.815 | 261.445 | 134.936 |
| $T_{lag}$ (h) | 3.640 | 4.344 | 3.637 | 5.603 | 0.149 | 0.419 |
| $R^2$ | 0.9373 | 0.9384 | 0.9373 | 0.9442 | 0.9771 | 0.9754 |
| **Higuchi with $F_0$: F = $F_0$ + $k_H$*t^0.5** | | | | | | |
| $k_H$ | 62.987 | 67.568 | 88.685 | 276.060 | 271.562 | 142.595 |
| $F_0$ | - | - | - | - | - | - |
| $R^2$ | 0.9788 | 0.9788 | 0.9788 | 0.9788 | 0.9788 | 0.9788 |

The different values for TX highlight that every polymer-based film presents particularities, depending on pH and the composition of the polymer matrix. At pH 5.5, flux varied between 8.058 $\pm$ 0.125 $\mu$g·cm$^{-2}$·h$^{-1}$ and 10.850 $\pm$ 0.380 $\mu$g·cm$^{-2}$·h$^{-1}$, and a direct proportioned increase was observed with the increase of the viscosity of the film-forming polymer. At pH 7.4, the TX flux varied between 10.990 $\pm$ 0.2.490 $\mu$g·cm$^{-2}$·h$^{-1}$ and 53.140 $\pm$ 0.196 $\mu$g·cm$^{-2}$·h$^{-1}$, showing a different behavior compared to pH 5.5. The flux decreased in parallel with the increase of the polymer viscosity and concentration (flow: $TX_1 > TX_2 > TX_3$).

Data from Table 3 suggests that under certain experimental conditions at pH 5.5, formulas $TX_1$ and $TX_3$ reached a steady state in 8 h, and maintained it for 22 h. Formula $TX_2$ reached a steady-state in 8 h as well, but maintained it for only 16 h. At pH 7.4, steady state for formulation $TX_1$ was reached in 6 h and remained constant for another 4 h. Formulas $TX_2$ and $TX_3$, based on high-viscosity polymer, were allowed to reach a steady state in 4 h, which they maintained for another 4 h. These data indicate that through kinetic parameter optimization calculated using the Higuchi kinetic function, there is a possibility of an enhancement of the flux through the membrane, but also a decrease in the time necessary to achieve the equilibrium state, as well as an increased period of time in order to be maintained.

### 3.4.2. Estimation of Kinetics for TX Dissolution from Polymeric Films

The Korsmeyer–Peppas model considers diffusion coefficient *n*. Exponent *n* is an indicator of the drug-dissolution mechanism: *n* = 0.5—Fickian transport mechanism; 0.5 < *n* < 1.0—non-Fickian transport mechanism (anormal transport); *n* = 1—case II transport mechanism; *n* > 1—super case II transport mechanism [39,40].

Using the obtained data (Table 4 and Figure 7), it may be deduced that the Korsmeyer–Peppas model describes a non-Fickian transport mechanism. The $n$ values varied between 0.63–0.7 at pH 5.5 and 0.73–0.86 at pH 7.4, suggesting a diffusion depending on the matrix hydration and polymer relaxation. To analyze the diffusion of TX from the matrix, the dissolution-curve fitting was realized in the period of a maximum of 60% of the initial concentration of TX dissolved. Taking into consideration the differences in experimental conditions for the discrimination of models, the Akaike information criterion (AIC) was used. The Akaike index is a "goodness of fit" type indicator of the best fit model possessing the lowest AIC value (Table 4).

**Table 4.** The kinetic parameters "best-fit values" and "goodness of fit" for the dissolution profiles of the polymeric film with TX.

| pH | pH 5.5 | | | pH 7.4 | | |
|---|---|---|---|---|---|---|
| **Formula** | **TX$_1$** | **TX$_2$** | **TX$_3$** | **TX$_1$** | **TX$_2$** | **TX$_3$** |
| | **Korsmeyer–Peppas**: $F = k_{KP}*t\char94 n$ | | | | | |
| **kKP** | 0.44 | 0.31 | 0.32 | 2.18 | 2.90 | 4.00 |
| $n$ | 0.63 | 0.69 | 0.70 | 0.73 | 0.86 | 0.74 |
| | **Korsmeyer–Peppas with T$_{lag}$**: $F = k_{KP}*(t-T_{lag})\char94 n$ | | | | | |
| **kKP** | 1.20 | 1.17 | 2.37 | 4.26 | 41.11 | 15.96 |
| $n$ | 0.83 | 0.86 | 0.73 | 0.89 | 0.27 | 0.36 |
| **T$_{lag}$** | −0.02 | 0.64 | 1.03 | 1.85 | 1.96 | 1.15 |
| **AIC** | −3 | −2 | 5 | 42 | 58 | 44 |
| | **Korsmeyer–Peppas with F$_0$**: $F = F_0 + k_{KP}*t\char94 n$ | | | | | |
| **kKP** | 1.20 | 0.98 | 1.80 | 2.40 | 26.96 | 13.04 |
| $n$ | 0.83 | 0.91 | 0.81 | 1.04 | 0.40 | 0.42 |
| **F$_0$** | 0.00 | 0.00 | 0.00 | 0.00 | 0.00 | 0.00 |
| **AIC** | −3 | 1 | 12 | 46 | 67 | 45 |

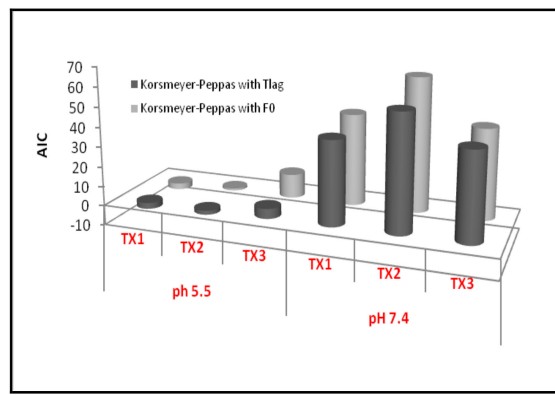
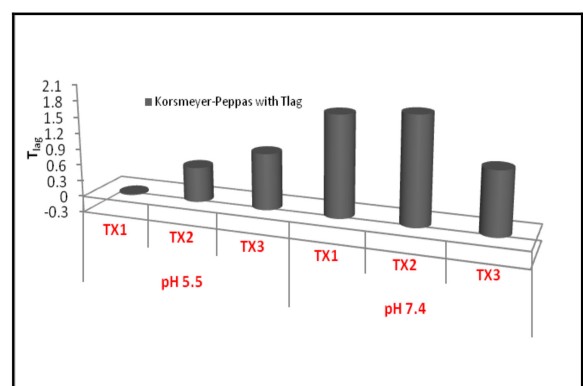
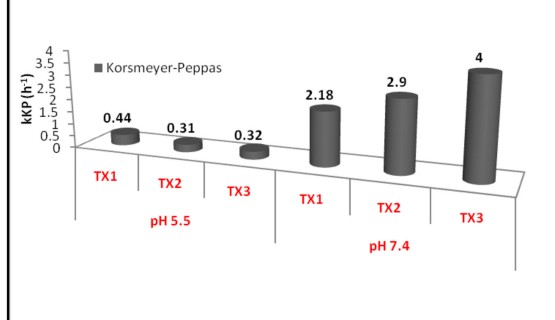
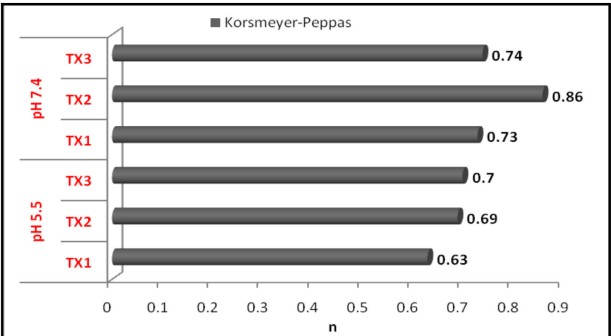

**Figure 7.** The parameters of the Korsmeyer–Peppas model.

Comparative analyses of the fitted data of Korsmeyer–Peppas with $T_{lag}$ and Korsmeyer–Peppas with $F_0$ models revealed that function with $T_{lag}$ is the best fit model, as its AIC values were low. Table 4 and Figure 7 show that at pH 5.5, latency time depends on the viscosity and concentration of the used polymer. Similar behavior can be observed at pH 7.4 as well. Formulation $TX_1$, based on low-viscosity HPMC, has shorter latency times compared to high-viscosity HPMC-based $TX_2$. Increasing the concentration of $HPMC_{15kcP}$ by 0.5% in $TX_3$ compared to $TX_2$ determined an increased quantity of the diffused active substance to the matrix surface. However, the rate constant $k_{KP}$ from the Korsmeyer–Peppas equation depends on both the formulation variables and experimental conditions of the in vitro drug release. All three TX films presented a significant increase in the release rate at pH 7.4 versus pH 5.5.

### 4. Conclusions

In this study, we proposed the formulation and evaluation of three polymeric films containing TX. The results of the DSC and FTIR analyses demonstrated the compatibility of the active substance with the two polymer types using cellulose ethers as film-forming substances. FT-IR spectra confirmed this result through the lack of significant modification in characteristic absorption bands. The DSC curves of binary mixtures indicated the same thermal behavior as a pure drug.

The AUC values indicated a major influence of acceptor-medium pH on the dissolution rate of the TX from the prepared polymeric films. Different flux values highlighted that each polymer film had a particular behavior, dependent on the pH and matrix composition. At pH 5.5, flux values were between $8.058 \pm 0.125$ $\mu g \cdot cm^{-2} \cdot h^{-1}$ and $10.850 \pm 0.380$ $\mu g \cdot cm^{-2} \cdot h^{-1}$, and at pH 7.4, flux values were between $10.990 \pm 0.2.490$ $\mu g \cdot cm^{-2} \cdot h^{-1}$ and $53.140 \pm 0.196$ $\mu g \cdot cm^{-2} \cdot h^{-1}$. Modeling the releasing curves using Higuchi kinetics indicated that through the optimized parameters there is a possibility of increasing the flux through the membrane, together with a reduction of the time needed to achieve the equilibrium state and an enlargement of the maintained duration. Mathematical modeling of dissolution curves using Korsmeyer–Peppas described a non-Fickian mechanism, with *n* values varying between 0.63–0.7 at pH 5.5, and 0.73–0.86 at pH 7.4.

**Author Contributions:** Conceptualization, A.C., P.A., N.T.; methodology, E.R., R.A.V., A.T., M.B., P.A.; software, N.T., P.A.; validation, A.C., N.T., P.A.; writing—original draft preparation, P.A., A.C.; writing—review and editing, P.A., N.T., A.C.; visualization, D.-L.M., A.C.; supervision, A.C. All authors have read and agreed to the published version of the manuscript.

**Funding:** This work was supported by the University of Medicine, Pharmacy, Science and Technology "George Emil Palade" of Targu Mures, Research Grant no. 275/6/11.01.2017.

**Data Availability Statement:** Data is contained within the article.

**Conflicts of Interest:** The authors declare no conflict of interest.

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
