# Peer review of "Polymeric Films Containing Tenoxicam as Prospective Transdermal Drug Delivery Systems: Design and Characterization"

_processes, doi:10.3390/pr9010136_

Round 1
Reviewer 1 Report
Most sentences in Abstract should be written in past tense.
Line 22: The sentence of “Widely used screening methods include FTIR and DSC” should be omitted.
Line 24: Three polymeric films: Three polymeric films of
Line 140: homogenized → mixed (?)
FTIR study (Figures 1 & 2): The authors reach a conclusion that there is no interaction between TX and polymers. It seems to be a too subjective interpretation. There should be also FTIR spectra for each polymer alone.
DSC study (Figure 3): The authors conclude that the binary mixtures of TX and 2 polymers indicate the characteristics comportment of TX. However, the relevant thermograms indicate that there is strong interactions between TX and 2 polymers, so that the thermal behavior of TX changes once it is mixed with each polymer. Also, DSC thermograms for each polymer alone should be provided.
Figure 4, TX release study: TX is practically insoluble in water. Yet, the authors used phosphate buffers at pH 5.5 and 7.4 as dissolution media. Do these aqueous solutions provide a sink condition for TX during the dissolution study?
Reviewer 2 Report
This manuscript studied the in vitro evaluation of tenoxicam-loaded polymeric films for potential transdermal delivery application. It is a well-written and well-organized manuscript. It is a well-attempted manuscript however including some additional info would be helpful to the readers.
Here are some comments:
- Section 2.2: FTIR studies were conducted only with a physical mixture (ambient temperature?) Since for the film preparation, drug/polymer solution was heated for 24 hours at 40C to dry the film, it would be helpful to run FTIR with the final formulation (Film) as well in addition to just the physical mixture to get a better picture of the interactions.
- Section 2.2.3. Table 1, Composition of the Polymeric Films. Since the composition listed is of the original solution and not the final dried polymeric film, it would be helpful to include the final targeted formulation composition post-manufacturing/processing (the actual composition of the film).
- Compositional characterization of the film post-manufacturing is missing (such as assay)
- Section 2.2.4: For dissolution studies, 1.8 cm diameter film was used, it would be helpful to include the weight of the sample and the amount of drug present in the sample (to understand the drug loading that could help understand some rate kinetics).
- Section 3.2 DSC analysis: Exothermic peak was seen in control and polymeric/drug samples, but the endothermic peak is present in drug control but is missing in the test (polymer/drug) samples, please elaborate? Have the authors ran the DSC of the final polymeric film? Including this data could be helpful.
- Additional explanation/potential theories from a chemistry perspective would be helpful for seeing the differences in the release kinetics at different pH for different polymeric films [%drug released at last time point (at pH 5.5 TX1:~20%, TX2: 20%, TX3: 27%), whereas at pH 7.4 TX1: ~80%, TX2:~96%, TX3: 55%)
Reviewer 3 Report
Authors in the article entitled "Polymeric films containing tenoxicam as prospective transdermal drug delivery systems: design and characterization" present the development and characterization of polymeric films with a possible application as TDDS. Below presented comments should help to improve the manuscript and solve some misunderstandings in the presented work.
1. There should be provided information about the solubility of TX in dissolution media applied in research for a better understanding of obtained differences in drug dissolution in different pH.
2. Why did the authors decide to use a dissolution medium with pH 7.4 for transdermal dosage form? It is quite far from the physiological pH of the skin.
3. Did authors make FTIR spectra for both HPMC used as excipients? It could be advised to add them to the manuscript even on one additional plot using two different colors to have a possibility to compare if Drug-Excipient FTIR spectra are simple average or some interesting changes occurs.
4. What was the dose of the drug in the film used for the dissolution study?
5. Lines 67-72 should be moved out from introduction to materials and methods. In general, the introduction should be reformated and a description of analytical methods should be moved to propper subsections of materials and methods. In the introduction, authors should provide additional information about TX use and recent advances in the field of transdermal route of administration especially in terms of polymeric films.
6. I don't understand the reasoning behind lines 207-211. A factor that will limit dissolutions and absorption is pH of the skin (around 4-6) so how authors assume that better dissolution in pH 7.4 will help to achieve therapeutic concentration? Moreover, we don't know the dose in the used sample.
7. Conclusions in lines 294-296 are wrong if authors think about TDDS.
Round 2
Reviewer 1 Report
The authors seem to do their best to adequately address the concerns raised by the reviewer. Please let me pinpoint one point that if a dissolution medium does not guarantee a sink condition, there is no point of performing an in vitro dissolution test.
Reviewer 2 Report
The authors tried to address earlier comments and I do not have additional comments.
Reviewer 3 Report
Thank you for your answers and the modifications you made in the manuscript. Now, I understand better the whole design and results and I believe that the article will find an audience in Processes Journal.